# Management of Thermal Injuries in Donkeys: A Case Report

**DOI:** 10.3390/ani10112131

**Published:** 2020-11-17

**Authors:** Jorge Lohse, Pierpaolo Pietrantoni, Christian Tummers

**Affiliations:** 1Equine Sport Medicine Advisor, Equinos Chile, Chorrillos, Viña del Mar 399, Chile; 2School of Agricultural Sciences and Veterinary Medicine, Viña del Mar University, Agua Santa, Viña del Mar 7055, Chile; ppietrantoni@uvm.cl; 3Natural Resources Faculty, Department of Veterinary Medicine and Public Health, Temuco Catholic University, Manuel Montt, Temuco 056, Chile; ctuemmers@uct.cl

**Keywords:** donkeys, thermal injurie, burns, fire

## Abstract

**Simple Summary:**

Reports or descriptions of medical management of thermal injuries in donkeys is lacking. Four donkeys sustained burn injuries during the wildland–urban interface fire in Valparaiso, Chile, in 2014. The donkeys received first aid care at the scene of the fire, and then were hospitalized and treated for three months for thermal injuries, of various degrees of severity, in several body parts. The clinical findings and treatment of four of the donkeys are described in this paper. All donkeys recovered and were retired to an animal shelter.

**Abstract:**

Burn injuries are uncommon in large animals and there are no reports of these injuries in donkeys. Burns cause local and systemic effects. Extensive thermal injuries can be challenging to manage and the extent of the burn surface affected will directly impact the severity of the illness and the prognosis. Burns are classified according to the depth of injury into four categories, from first-degree burns, and the least affect to fourth-degree burns, which are the more severely affected patients. This case report describes the medical management of four donkeys that sustained various degrees of external burn injuries during the wildland–urban interface fire in Valparaiso, Chile. The donkeys were treated topically for several weeks and closely monitor for inadequate nutritional intake. Water based topical medications are preferred in burn cases because they can be easily applied and removed without interfering with wound healing. Of note, the caloric demands of these cases can be achieved by increasing the amount of grain, adding fat (i.e., vegetable oil), and free-choice alfalfa hay. All donkeys recovered and were retired to an animal shelter.

## 1. Introduction

Burns are damage to body tissues caused by the action of high temperature or hot solutions, lightning, friction, abrasions, or certain chemicals (alkalis, acids, salts of heavy metals) [1]. Burns are uncommon in horses and there are no reports of thermal injuries in donkeys in scientific literature. Emergency treatment guidelines are extrapolated from human and equine medical literature [2,3,4,5,6].

From 12 to 14 April 2014, a wildland-urban interface (WUI) fire occurred at a catastrophic scale. Over 2827 acres were burned, 11,000 people affected, and 2900 homes destroyed in Valparaiso, Chile. WUI fires occur more frequently during periods of drought and climate change. The wildfires also significantly affect land used for agriculture and wildlife habitats, injuring and killing large numbers of wildlife and farm animals. Valparaiso has a peculiar topography characterized by 45 hills, and the risk for wildfires exist in most parts of rural areas every summer. The city of Valparaiso was declared a World Heritage Site by United Nations Educational, Scientific and Cultural Organization (UNESCO) in 2003. The estimated number of affected domestic animals associated with the WUI fire was 146 (70 horses, 33 donkeys, 29 goats, 11 pigs, 2 cows, and 1 sheep), of which many of them died. Many of those animals were examined and treated by veterinarians for burns and/or smoke inhalation. Four donkeys suffered burns, two of them severe, requiring long-term hospitalization, and costly medical treatment, therefore, the owner donated the donkeys to the Veterinary Clinic Service of Viña del Mar University.

During fires, thermal injuries are caused by direct exposition to the flames and/or inhalation of toxic gases [7]. Inhalation injury is a common sequel of closed-space fires and develops through three mechanisms: direct thermal injury, carbon monoxide poisoning, and chemical insult [4]. Horses exposed can suffer respiratory injury of varying degrees, ranging from mild irritation to severe smoke inhalation-induced airway or lung damage [8]. The extent of the burns can vary in terms of the surface area affected and the depth of tissue injury. Thermal injury can be life-threatening due to subsequent denaturation of cellular metabolic processes and apoptosis, leading to necrosis [9].

Classification of the burn injuries is considered one of the more important determinants of outcome. First-degree burns affect only superficial layers of the epidermis, causing erythema, edema, and desquamation, and they heal without complication. Second-degree burns involve the superficial and/or deep portion of the epidermis. Superficial second-degree burns affect the stratum corneum, stratum granulosum, and a few cells of the basal layer; however, deeper burns involve the epidermis, including the basal layers. Such burns are characterized by erythema and edema at the epidermal-dermal junction, necrosis of the epidermis, accumulation of leukocytes at the basal layer of the burn, eschar formation, and minimal pain [10]. These may heal within 3 to 4 weeks if further dermal ischemia is prevented. Third-degree burns affect all layers of the epidermis and dermis, including the adnexa. These burns heal by contraction and epithelialization from the wound margins and secondary infections are common complications. Fourth-degree burns affect the skin and underlying muscle, bone, ligaments, fat, and fascia [10,11,12].

In addition to the depth of the burn, the extent or total body surface area (TBSA) affected by thermal injuries also needs to be assess. The “Wallace Rule of Nines” system used in human medicine estimates the prognosis according to the extent of the burn. This system divides the body surface area into regions that represent multiples of nine and allocates body regions as follows: each limb, head, neck, thorax, and abdomen represent 9% [13]. Specific guidelines of the TBSA do not exist in large animals [10,11,12]. Other evaluation methods are described [12].

Donkeys are particularly susceptible to hyperlipemia in stressful circumstances. It can progress rapidly and is often life-threatening. Prompt diagnosis and treatment is required to improve the outcome. If managed correctly, with an appropriate diet and regular (routine) preventive healthcare, this risk can be minimized [14].

This report describes the medical management of veterinary care of thermal injuries in four donkeys.

## 2. Materials and Methods

### History and Case Presentation

Case 1 (D1): a 7-year-old male donkey, weighing 150 kg was examined and treated within 2 h after being burned. The donkey was kept in a small pen (10 × 10 m) with three other donkeys. Upon physical examination, the body condition score (BCS) was 3/5, tachycardic 64 beats/min (reference range 36–52), tachypneic 30 breath/min (reference range 12–28), furthermore, the donkey had congested mucous membranes. There were burn marks over the lips, face, ears, dorsum, and both front limbs. All serous exudate was observed over the coronary bands in front limbs, but had a normal gait. Immediate treatment included flunixin meglumine 1.1 mg/kg intravenously (IV) (83 mg/mL, Febrectal, Drag Pharma Laboratory Chile Invetec S.A., Santiago, Chile); hosing with tap water (13 o 15°C) all over the body for 30 min to protect the tissue, and bandage with sterilized gauze and liquid paraffin gauze dressing over the neck and back prior to transportation to the clinic 2 h later (Figure 1A).

Approximately 12 h after the incident, the estimated TBSA was approximately 40–50%. Most of the affected areas were classified as deep second-degree burns with some small areas of third-degree burns on the distal area of the front limbs and face. There was also facial, abdominal, and limb edema. The donkey was eating and drinking with difficulty (dysphagia—possibly caused by edema and pain). He was placed on IV fluids with Hartmann’s solution 10 mL/kg bwt/h (Lactated Ringer’s solution, Baxter, Toongabbie, New South Wales, Australia) for 3 h until he was stable and started to eat and drink. Both eyes had bilateral corneal ulcers and uveitis (Figure 1B), with signs, such as epiphora, blepharospasm, corneal edema, and episcleral congestion. Treatment with ophthalmic antibiotic ointment containing zinc bacitracin 400 iu/g, neomycin sulfate 3.5 mg/g, and polymyxin B sulfate 6000 IU/g q 12 h (Oftabiotico, Saval S.A. Laboratory, Santiago, Chile), ophthalmic atropine sulfate drops q 12 h × 2 days (10 mg /mL, Atropina 1%, Saval S.A. Laboratory, Santiago, Chile), and dextran-70 lubricant drops 1 mg with hidroxipropilmetilcelulosa 3 mg (Tears Naturale II, Alcon Laboratory Chile Ltd.a., Santiago, Chile) was initiated several times a day. The eyes were treated with triple antibiotic ointment for six weeks, and natural tear drops for four weeks.

A topical cream of sulfanilamide with zinc oxide was applied to the coronary bands (sulfanilamide 5 g, zinc oxide 5 g, vitamin A 250,000 IU, vitamin D2 90,000 IU/100 g, Pomada Sulfavitaminada, Drag Pharma Laboratory Chile Invetec S.A., Santiago, Chile).

Respiratory distress was observed, characterized by tachypnea, nostril flaring, and cough—a suspected result from smoke inhalation; therefore, treatment with corticosteroid, dexamethasone sodium phosphate 0.2 mg/kg IV (5,26 mg/mL, Hasyun, Drag Pharma Laboratory Chile Invetec S.A., Santiago, Chile) was administered. A parasympatholytic drug and beta-2 agonist inhalation therapy by aerochamber (Aerofacidose Aerocamara A.E., Chile Laboratory S.A. Teva, Santiago, Chile), using ipratropium bromide 0.8 µg/kg q 6 h (250 µg/mL, Atrovent, Boehringer Ingelheim International GmbH, Rhein, Germany) and albuterol 2 µg/kg q 4 h (100 µg/doses, Sinasmal SF Salbutamol, Chile Laboratory S.A. Teva, Santiago, Chile) were also administered. The aerochamber was placed in one nostril and the opposite side was occluded (Figure 2). Airway endoscopy was performed the second day (1800PVS9150, PortaScope, FL, USA), and mild to moderate irritation with hyperemia of the mucous membrane from a common nasal meatus, and the first part of a ventral nasal meatus of both nostrils, as well as mild hyperemia in the pharynx and trachea, were observed. The cough resolved within 48 h of treatment; therefore, the bronchodilators were discontinued after three days.

Complete blood cell count and serum biochemistry profile were performed on the second and eight days post rescue (Biosystems BA400, Biosystems S.A., Barcelona, Spain). The first analysis, done at day two, revealed: neutrophilia, increased alkaline phosphatase, muscle enzymes creatine kinase, aspartate aminotransferase and triglycerides, decreased total protein and albumin, increased glucose, and decreased total calcium and chloride concentration. Slightly cloudy serum was observed in the clotted blood sample (Table 1).

The second analysis at day 8 showed increased alkaline phosphatase, aspartate aminotransferase. Blood analysis was repeated on day 15 of the hospitalization and all of the parameters were within normal limits (Table 1). Antibiotics therapy was initiated with procaine g penicillin 20,000 IU/kg + dihydrostreptomycin 15 mg/kg q 24 h intramuscular (IM)/7 days (200,000 IU procaine G penicillin + 250 mg dihydrostreptomycin sulfate/mL, (Pentril, Drag Pharma Laboratory Chile Invetec S.A., Santiago, Chile).

During the hospitalization time period (90 days), the burn surfaces, appetite, and pain level of the donkey were monitored. The donkey appeared more comfortable, with good appetite, and showed interest in socializing with other donkeys and his environment; these responses became evident since day 7 in the clinic. The edema (described) was observed since the first day of the rescue and decreased after 6 days, the skin became dry and sensitive to touch, and he experienced hair loss on his legs, neck, and back.

Flunixin was given daily for pain control for 10 days since the rescue (3 days, 1.1 mg/kg q12 h, then 7 days, 1.1 mg/kg q 24 h IV), and continued with a natural product (devil’s claw 726 mg + *Yucca schidigera* extract 954 mg + vitamin B12 10 mcg/ 10 mL, Bute-Less Paste, Absorbine, Massachusetts, MA, USA), 5 mL oraly q 24 h/15 days, supplement to discomfort and pain relief. Omeprazole 2 mg/kg per os q 24 h/4 weeks (20 mg tablets, Omeprazole, Chile Laboratory S.A. Teva, Santiago, Chile) was given for treatment and/or prevention of gastric ulceration.

During the following 2 weeks, the eyelids, ears, and back became dry, hairless, and painful, and there was a loss of elasticity (Figure 3). Topical treatment with aloe vera gel was initiated twice daily to exposed areas before washing and debridement. The pinna of both ears developed ulcers and contracture of the cartilage.

Daily skin care initially required sedation xylazine 0.5 mg/kg IV, with butorphanol tartrate 0.01 mg/kg IV (10 mg/ mL, Torbugesic, Zoetis, NJ, USA) as potent analgesic for the first six days in the clinic. On day seven, sedation and potent analgesic for treatment was no longer required. Local anesthetic instillation with lidocaine hydrochloride was used in some affected areas on the back and face (20 mg/ 1 mL, Lidocalm, Drag Pharma Laboratory Chile Invetec S.A., Santiago, Chile).

Wounds were lavaged every 24–36 h, for debridement of devitalized tissue, and dead skin was removed with sterile scissors or a hypodermic needle. The blisters were left intact, however, gauze dressing with impregnated paraffin (Jelonet) was used in some areas where blisters opened. Local antibacterial drugs were used every 24 h. The wounds were lavaged with chlorhexidine gluconate solution (2 g/100 mL, Inveclor, Drag Pharma Laboratory Chile Invetec S.A., Santiago, Chile) and diluted in water with sterilized gauze using gloves and triclosan (0.3%/100 g, Sanigermin, Sanitas S.A. Laboratory, Santiago, Chile). A variety of topical products were used in conjunction with the dressings for the following weeks: cream of *Buddleja globosa* (Matihorse, Drag Pharma Laboratory Chile Invetec S.A., Santiago, Chile), raw honey, aloe vera, and nitrofurazone cream (0.2 g/100 g, Furasep Drag Pharma Laboratory Chile Invetec S.A., Santiago, Chile). Nitrofurazone was used in areas with exudation and erythema and, locally, raw honey on areas with less exudation and erythema. Dry skin areas were treated with Jelonet, and then it was changed to *Buddleja globosa* cream. Fly repellent was used around wounds (dichlorvos 1250 mg + triclosan, 500 mg/ 100 g, Moskation larvicida spray, Drag Pharma Laboratory Chile Invetec S.A., Santiago, Chile).

Three weeks later, the donkey developed pruritus, which was treated with chlorpheniramine maleate 0.5 mg/kg IV (50 mg/mL, Histamil, Troy Laboratories Pty. Limited, Glendenning, Australia) as needed.

After 2 months, a thin layer of skin started to regrow. Exuberant granulation tissue on some areas was periodically de-bulked. After 3 months, all treatments were discontinued, and most of the burned areas had healed, but the area of the face and front limbs were without hair growth. At this time, the only product used on his legs and body was aloe vera gel. His coronary bands from the front limbs did not separate significantly, but a horizontal line of damage was evident around the hoof wall (1 cm below the coronary band), walk, and trot sound.

Case 2 (D2): a 15-year-old female donkey, 120 kg, was examined with D1. The jenny appeared anxious and painful. Upon physical examination, her BCS was 3/5, she was tachycardic (72 beats/min), tachypneic (30 breath/min), and had congested mucous membranes and muscle tremors. Burns over her face, neck, back, perineal area, vulva, and coronary band swelling of the left hind limb were observed. She was transported to the clinic with the other affected animals and was treated, as per case 1.

Her burns were classified as second-degree, with some areas with third-degree (perineal area); estimated TBSA approximately 40–50%. She had bilateral corneal ulcers and uveitis, and her eyes were treated with the same antibiotic ointments and regime as case 1. The left coronary band was inflamed and a serious exudate was present; therefore, topic cream with sulfanilamide + zinc oxide was applied. She was five-months in gestation, which was closely monitored at the clinic. Intravenous fluids with Hartmann’s solution were administered at the same rate as described above, as well as flunixin and omeprazole. Fluids were discontinued on the second day at the clinic, when the laboratory report indicated adequate hydration. Moreover, she was eating hay with a good appetite at this time.

Complete blood cell count and serum biochemistry profile were performed in the clinic similar to D1. The first blood analysis, taken at day two, revealed: neutrophilia, increased alkaline phosphatase, muscle enzymes creatine kinase, aspartate aminotransferase, triglycerides and cholesterol, decreased total protein and albumin, increased glucose, and decreased mineral calcium and chloride concentration. Slightly cloudy serum was observed in the clotted blood sample (Table 1).

Antibiotic therapy started with procaine penicillin + dihydrostreptomycin IM, and for the skin, aloe vera cream, at the same doses rates described in D1. The wounds from the skin were periodically lavaged and cleaned using disinfectant, and a similar topic cream, as described in D1, removing the devitalized tissue when necessary (sedation and potent analgesic, described in D1, were previously used). The pinna of the left ear also developed ulcers and permanent cartilage contracture, as D1. A second blood analysis performed eight days post rescue showed increases in alkaline phosphatase and aspartate aminotransferase. Concerning the third blood analysis performed on the 15th day of hospitalization, all parameters were within normal limits (Table 1). Pruritus was evident on day 10; therefore, the antihistamine product used was the same as D1. Over the next 2 weeks, her wounds continued to improve and the burns re-epithelialized rapidly (Figure 4).

The left coronary band lesion, increasing in size and lameness, was graded 2/5. Radiographs were taken, which revealed no sign of distal phalangeal rotation. Flunixin meglumine (1.1 mg/kg) for pain was continued, daily, for 12 days (3 days 1.1 mg/kg q12 h, then 9 days 1.1 mg/kg q 24 h IV), and the supplement for pain relief was the same dose rates as D1. After six weeks, she could canter when turned out into a paddock in the evenings, or early mornings, when the outdoor temperatures were cooler. His left coronary band from the hind limb did not separate, but a horizontal line of damage was evident around the hoof wall (1 cm below the coronary band).The jenny was discharged after 3 months of hospitalization. She was doing well on her 3- and 8-month pregnancy checks. She foaled a healthy donkey foal after 13 months of gestation (range of 11–14.5 months) [11].

Cases 3 and 4 (D3–D4): the next two cases were a 12-year-old female donkey, 160 kg, and a 5-year-old male, 110 kg. Both were examined at the site of the fire, as described for both previous cases (D1 and D2). They both had a BCS of 2/5, were tachycardic, and small burn areas with hair loss on their heads (noses and cheeks), as well as necks, were observed. Their TBSAs were approximately 5% and the burn areas classified as first degree. These two donkeys were in the same paddock with the others; however, they were less affected. Cases D1–D2 were tied up to a fence, close to the burning vegetation until the rope was burned by fire, while cases 3 and 4 were free. This most likely explains the severity of the injuries in the first two cases.

D3 and D4 were also hosed with tap water all over their bodies, and moved to the same clinic. They were also treated with flunixin meglumine for 3 days and the burn areas were treated with aloe vera as the previous two cases. Superficial burns healed without complication. The areas affected became dry between 5 and 8 days. The wounds from the skin were periodically lavaged and cleaned using disinfectant, and similar topic cream, as described in D1, was used. It was necessary to use xylazine and butorphanol tartrate for wound management for 2 days after they arrived at the clinic. After 3 weeks, most of the burned areas healed (Figure 5).

The hematologic and biochemical analyses of D3–D4 performed two days after the recue showed no abnormalities (Table 2). They showed no signs of discomfort and had good appetite. They were treated for 5 days and then discharged from the clinic to a free ranged pasture.

All of the donkeys were fed alfalfa hay and grass (*Medicago sativa* and *Lolium perenne*), 1.7% daily of their bwt in dry matter (DM) intake, which is equivalent to 2.6 kg D1, 2.1 kg D2, 2.7 kg D3, 1.9 kg D4 per day (Figure 6). What was also given daily included: vegetable oil, 30 mL, and 20 mL of multivitamin and mineral oral supplements for horses containing: copper 150 mg, cobalt 6 mg, potassium 330 mg, magnesium 70 mg, manganese 130 mg, zinc 370 mg, iron 1000 mg, calcium pantothenate 170 mg, folic acid 35 mg, biotin 0.08 mg, vitamin: A 80,000 IU, B1 200 mg, B2 100 mg, B6 35 mg, B12 400 µg, D2 12,000 IU, E 150 IU, sucrose 50 G, choline chloride 675 mg, and selenium 3.1 mg/ 100 mL (Equifort, Drag Pharma Laboratory Chile Invetec S.A., Santiago, Chile). All were orally de-wormed with fenbendazole 7.5 mg/ mL (10 g/100 mL, Lombrimic, Drag Pharma Laboratory Chile Invetec S.A., Santiago, Chile).

## 3. Results and Discussion

To the best of our knowledge, this is the first case report on the medical management of severely burned donkeys. There is extensive literature regarding thermal injuries in humans and horses, and the lack of reports of thermal injuries in donkeys may be due to the costs associated with treatments, which are usually prolonged and expensive. The market value of donkeys is usually lower than horses, and owners cannot afford to cover the medical treatment. Therefore, euthanasia may be their only option. In these cases, the owner donated the donkeys to the attending veterinary clinic for treatments, and then, rehomed them to a shelter.

After first aid assessment and treatment at the site of the fire, the donkeys were transferred and fully evaluated at the clinic. The day after the fire, D1–D2 injuries were worse than initially assessed, and the extent of burns was greater than the predicted survival rate reported in the literature [15].

Furthermore, estimates of the extent and depth of the burns made prior to admission to a burn center, as well as experts in this field, have consistently shown to be inaccurate, despite standardization attempts and availability of tools, such as the Rule of Nines [9,16]. Given the unreliability of burn size and depth assessment by clinicians who are not burn experts, developing a scale for burn equid patients, we believe, is essential. A new approach from human medicine includes the use of computer-assisted programs to improve burn size estimation [17,18]. The treatment of donkeys with extensive burns was a major challenge. After the initial fluid therapy, however, the most severely affected donkeys (D1 and D2) improved and started to eat and drink. At this stage, however, prognosis for survival remained guarded.

Earlier intervention of burn patients is associated with favorable outcomes, lower complication, and mortality rates. The immediate emergency recommendation for severe burns is the application of copious amounts of cold water for a minimum of 20 min, within the first three hours of injury [19,20]. Water with temperature between 2 °C and 15 °C is preferred [20]. All donkeys described in this report, were managed using the considerations described (2 h after bring burned). Overcooling, resulting in hypothermia, was reported to be associated with adverse outcomes, including clotting disorders and increased mortality [19,20]. Animal studies showed that the immediate application of cold water was associated with faster re-epithelialization and reduced scarring [21].

D1 showed respiratory signs from smoke inhalation, which resolved in 3 days with the established treatment. The area where the donkeys were rescued from the fire that day was windy, which helped blow away smoke and ash into the atmosphere—this could be why D2 to D4 did not present problems in their respiratory systems. A case report of horses injured in open range fires described one horse with problems from smoke inhalation, which resolved in 24 h [5]. Barn fires are unfortunately too common and, each year, hundreds of horses die or are severely injured in these incidents. Gimenez et al. describe a review of strategies to prevent and respond to barn fires in the horse industry [22].

All of the parameters of complete blood count and serum biochemistry profile were normal, since the second analysis.

In some areas of deep second-degree and third-degree burns, skin grafting may be considered [4]; however, in these donkeys this intervention was unnecessary, since the wounds healed as per the reported time, according to the burn depth classification [4].

There are several methods to treat burn wounds in a horse, and the choice depends on the extent and location of the injury. Full-thickness burns can be managed by occlusive dressings (closed technique), continuous wet dressings (semi-open technique), eschar formation (exposed technique), or excision and grafting [4]. In this case report, bandaging the burned limbs was not performed as mentioned in burned horses [5], and was managed as open wounds with no complications. The use of the compression garments or compression suits post-burn injury is described as an important component of a human patient’s rehabilitation program, and reported for horses [5,23,24].

Cleaning with mild soap and water was used to facilitate softening and removal of dead skin. Aloe vera appeared to be helpful for managing dry skin, which appeared to prevent the skin from sloughing, and allegedly has anti-inflammatory effects and promotes wound healing [25,26]. This product had been reported to cause pruritus in some horses [5], but such adverse side effects were not observed in D1 and D2. The precise mechanism of post-burn pruritus has not been elucidated in humans or animals, but it appears to have pruritogenic and neuropathic aspects [27]. The occasional pruritus presented in D1 and D2 were managed by the use of chlorphenamine. Many burned equine patients are pruritic, and measures must be taken to prevent self-mutilation of the wound [4]. Reserpine had been used in burned horses for this complication, decreasing the urge to scratch by successfully breaking the itch–scratch cycle [4,5]. One author mentioned that the use of air conditioning fans directed to the stall may help [5]. The same author mentioned that topical corticosteroid creams were ineffective at controlling the itching in burned horses [5].

Honey has been used in burn patients for many years. The antibacterial activity of honey, its low pH, high viscosity, hygroscopic effect, and hydrogen peroxide content, may play a combined role in its effectiveness for the treatment of burns, and may also provide a moist environment for optimum healing conditions [28,29]. There are several types of honey—raw honey was used in these donkeys, and appeared to be useful in superficial and moderate burns, as reported in horses [5].

The donkeys with corneal ulcers (D1 and D2) responded well to treatment and healed with no complications. Human studies report 7.7%−18% of the incidence of chemical and thermal injuries to the eye of all ocular trauma, with corneal ulcers and blepharospasm [30]. Burned horse may have blepharospasm, epiphora, or both, which signify corneal damage [4]. Ocular emergency treatment by thermal injuries in humans is described [30]. Case reports in horses injured in open range fires that reported corneal ulcers were successfully treated with twice-daily treatments of topical antibiotics (zinc bacitracin, neomycin sulfate and polymyxin B sulfate), as well as daily cloxacillin benzathine eye ointment [5].

Long-term use of nonsteroidal anti-inflammatory drugs (NSAIDs) did not result in adverse effects during hospitalization, but other NSAIDs could have been used described for donkeys [14,31]. Gastrointestinal motility and appetite were normal in the first two days for donkeys 1 and 2. The prophylactic use of omeprazole may have been beneficial in preventing gastric ulceration with long-term use of NSAIDs. Omeprazole intended for use in humans was prescribed to these cases by the economic possibility of that moment, but there is no evidence about their efficacy in preventing ulcers in these donkeys. The authors recommend the use of an approved omeprazole product manufactured for horses, using the standard equine dosage appropriate for donkeys [14].

The application of sulfanilamide with zinc oxide around the coronary band was an effective topical treatment in these cases. Sulfanilamide acted with a differential bactericidal activity, and mild astringent, and decongestive effect, in addition to the absorbent and insulating properties conferred by zinc oxide [32,33]. Coronary band effusion was noted on the first day on D1 and D2, but separation was also noted. Abnormal hoof growth was observed—moved down over time until the old hoof was replaced—as was noted 8 months after the fire. The thickened hoof wall distal to the coronary band protected the majority of the laminae from thermal injury [5].

Preventive antibiotic therapy was used in D1 and D2, but the literature mentioned that the role of prophylactic antibiotics for severe burns is controversial, both in humans and horses [34]. Therefore, their use has not been advocated in recent guidelines or recommendations owing to a lack of evidence for efficacy and induction of antibiotic resistance. Systemic antibiotics do not favorably influence wound healing, fever, or mortality, and can encourage the emergence of resistant microorganisms. Additionally, circulation to the burned areas is often compromised, making it highly unlikely that parenteral administration of antibiotics can achieve therapeutic levels to the wound [4]. However, topical antimicrobials have been the mainstay of nonsurgical burn treatment [16]. In human reports, the use of prophylactic antibiotics may result in improved 28-day in-hospital mortality in mechanically ventilated patients with severe burns, but not in those who do not receive mechanical ventilation [35]. The use of antibiotics had been recommended in equine burn cases because of puncture wounds that occurred while attempting to escape the fire [5]. We do not find evidence that systemic therapy was useful, and we will not recommend it in future cases.

A hallmark of burn injury is a hypermetabolic response that results in significant pathological alterations in a number of tissues [36]. Studies in large animal models of burn hypermetabolism showed that 25% TBSA burns can generate a hypermetabolic response greater than smaller animals and closer to that seen in human patients [37,38]. It is recommended to gradually increase the grain, adding fat in the form of vegetable oil, and offering free-choice alfalfa hay increase caloric intake as nutritional needs for burned horses [4]. An anabolic steroid may be used to help restore a positive nitrogen balance [4]. Donkeys are highly efficient at digesting poor nutritional quality fiber and have lower energy requirements than horses and ponies of similar size (it is even lower in sick donkeys) [14,39,40]. Equine feedstuff based upon cereals or containing high levels of molasses should be avoided. It has been shown that they are risk factors for the development of gastric ulcers, laminitis, hyperlipidemia, and fatty liver disease in donkeys [14,40]. Early intervention to restore a positive energy balance, even before triglyceride values, are known to greatly increase the chance of survival. The restoration of a positive energy balance will stimulate endogenous insulin secretion and switch off lipolysis. Soya bean meal or alfalfa are excellent sources of digestible protein for convalescent donkeys [14,40]. The primary goal of this response is to provide adequate energy levels to maintain organ function and whole-body homeostasis. In addition to the food offered, multivitamin and mineral supplements, and vegetable oil to increase the energy intake, were added.

In veterinary emergency medicine and critical care, teamwork to provide adequate patient care is essential. The entire team consisted of two veterinarians, a technician, and eight graduate students to provide continues care. The care and devotion provided to each animal by the veterinarian staff highlights the concept of the animal-human bond and the dedication veterinarians provide to their patients.

## 4. Conclusions

Domestic and wild animals can suffer major thermal injuries during wildfires, and reports of these injuries in donkeys is lacking. This manuscript describes the clinical findings, medical treatments, and evolution of four donkeys that suffered thermal burns. All of the donkeys recovered and were sent to an animal shelter after their recovery.

## Figures and Tables

**Figure 1 animals-10-02131-f001:**
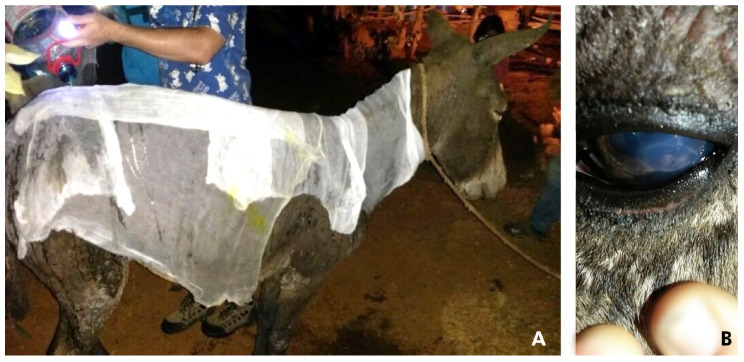
Donkey 1 received emergency medical attention for thermal injuries. (**A**) A large gauze dressing was applied over his body and soaked with tap water. (**B**): corneal ulcer.

**Figure 2 animals-10-02131-f002:**
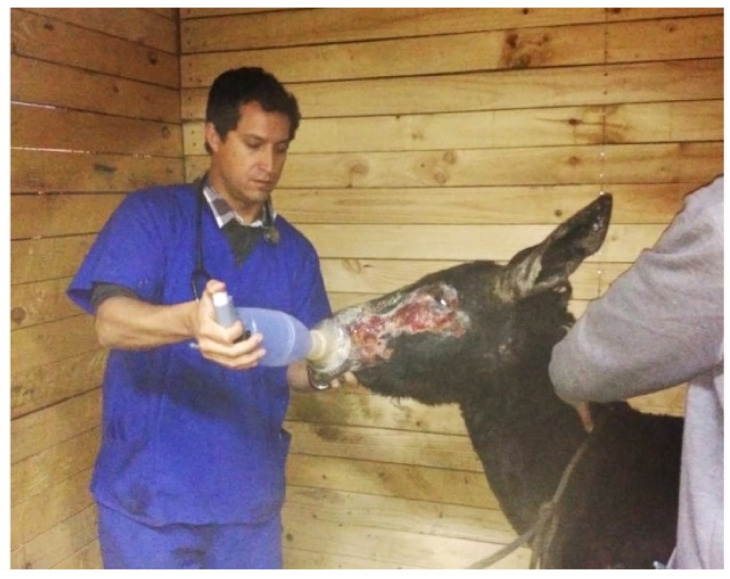
Use of the aerochamber for inhalator therapy in donkey 1, placed in the left nostril with the opposite side occluded.

**Figure 3 animals-10-02131-f003:**
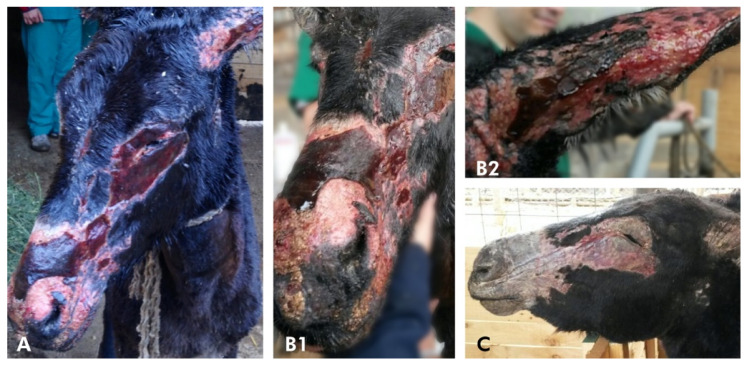
Donkey case 1, showing changes of the burned skin overtime. Appearance of the face after (**A**) after 5 days; (**B1**) 12 days; (**B2**) 12 days, external pinna area of the left ear; and (**C**) 20 days, respectively.

**Figure 4 animals-10-02131-f004:**
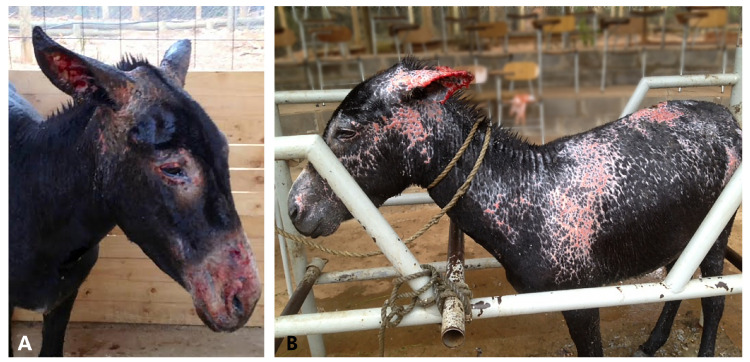
Donkey 2, showing changes of the burned skin over time. Appearance of her face and body at (**A**) 10 days; and (**B**) 20 days, respectively.

**Figure 5 animals-10-02131-f005:**
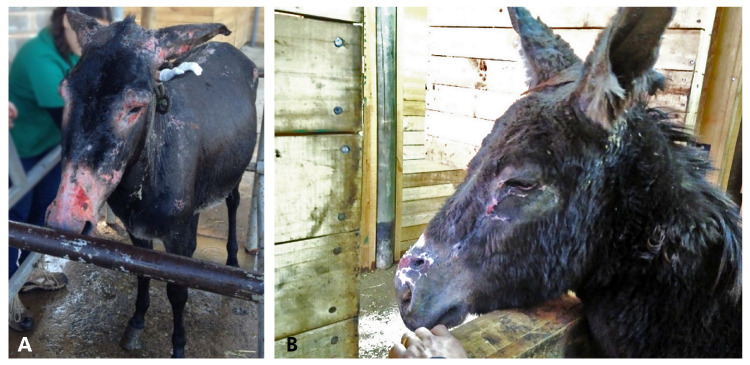
Appearance of burned skin in donkeys 3 (**A**) and 4 (**B**), 5 days after the fire.

**Figure 6 animals-10-02131-f006:**
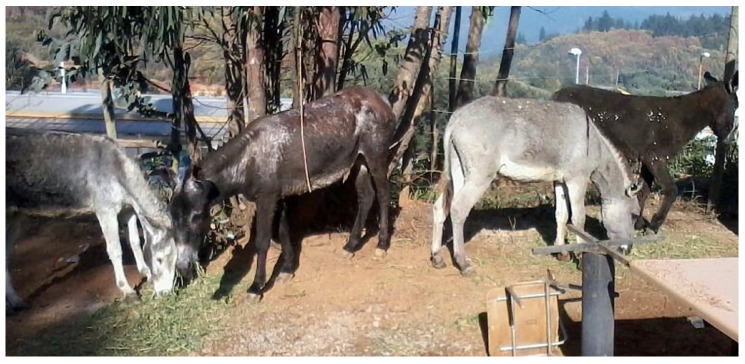
Donkeys D1–4 fully recovered, eating alfalfa hay and grass (*Medicago sativa* and *Lolium perenne*) 90 days after the fire. Left to right is D4, D2, D3, and D1.

**Table 1 animals-10-02131-t001:** Serial complete blood count and serum biochemistry profile of two burned donkeys (D1–D2).

Parameters	D1	D2	D1	D2	D1	D2	RI LRL-URL
	**Day 2**		**Day 8**		**Day 15**		
RBC (×106/µL)	4.6	5.5	6.6	6.6	6.8	7.0	4.4–7.1
Hemoglobin (g/dL)	11.1	9.3	11.4	11.4	12.2	13.9	8.9–14.7
Hematocrit (%)	30.0	29.0	32.0	36.0	32.0	38.0	27–42
WBC (×109/L)	15.1	15.8	14.8	14.2	11.1	13.4	6.2–15.0
S. neutrophils (×109/L)	8.4	7.1	6.2	6.0	6.0	5.5	2.4–6.3
Lymphocytes (×109/L)	4.1	1.5	1.9	3.0	2.6	2.7	2.9–9.6
Eosinophils (×109/L)	0.3	0.1	0	0,7	0.1	0.1	0.1–0.9
Monocytes (×109/L)	0.2	0	0.1	0	0	0	0–0.75
Thrombocyte (×109/L)	210	146	167	117	133	158	95–384
Urea (mmol/L)	4.0	3.0	4.2	3.6	1.9	2.5	1.5–5.2
Creat (µmol/L)	55	72	81	87	70	82	53–118
ALP (U/L)	328	293	291	270	123	201	98–252
AST (U/L)	829	712	621	589	399	512	238–536
GGT (U/L)	26	20	36	41	29	36	14–69
CK (U/L)	1024	619	319	375	326	208	128–525
Trig (mmol/L)	4.2	3.2	2.5	2.1	2.2	1.8	0.6–2.8
Total Proteins (g/L)	55	48	58	62	71	58	58–76
Albumin (g/L)	21	19	23	22	26	28	22–32
Globulin (g/L)	33	46	39	41	35	47	32–48
Tbil (µmol/L)	0.9	0.3	0.9	1.0	2.1	2.2	0.1–3.7
Chol (mmol/L)	2.9	3.0	2.5	2.8	1.9	2.3	1.4–2.9
Glucose (mmol/L)	5.1	4.0	4.5	4.6	4.1	4.5	3.9–4.7
Calcium (mmol/L)	2.1	1.9	2.5	2.9	3.0	3.2	2.2–3.4
Na (mmol/L)	130	131	129	135	129	133	128–138
K (mmol/L)	3.5	5.0	3.9	4.1	4.9	3.9	3.2–5.1
Cl (mmol/L)	95	100	98	104	99	104	96–106

D1: case donkey 1; D2: case donkey 2; RBC: Red Blood Cells; ALP: Alkaline phosphatase; AST: Aspartate aminotransferase; GGT: Gamma-glutamyl transpeptidase; Creat: Creatinine, BUN: blood urea nitrogen, CK: Creatine phosphokinase, Tri: Triglycerides, Tbil: Total bilirubin, Cho: Cholesterol, Na: Sodium, K: Potassium, Cl: Chloride. RI = Reference interval. LRL = Low RI. URL = Upper RL [14].

**Table 2 animals-10-02131-t002:** Complete blood count and serum biochemistry profile of two burned donkeys (D3–D4).

Parameters	D1	D2	RI LRL-URL
RBC (×10^6^/µL)	4.9	6.3	4.4–7.1
Hemoglobin (g/dL)	8.7	9.8	8.9–14.7
Hematocrit (%)	31	38	27–42
WBC (×10^9^/L)	7.4	14.2	6.2–15.0
S. neutrophils (×10^9^/L)	3.1	5.8	2.4–6.3
Lymphocytes (×10^9^/L)	3.5	6.9	2.9–9.6
Eosinophils (×109/L)	0.1	0.3	0.1–0.9
Monocytes (×109/L)	0.2	0	0–0.75
Thrombocyte (×109/L)	128	305	95–384
Urea (mmol/L)	4.0	3.6	1.5–5.2
Creat (µmol/L)	53	92	53–118
ALP (U/L)	104	211	98–252
AST (U/L)	257	322	238–536
GGT (U/L)	26	49	14–69
CK (U/L)	130	513	128–525
Tri (mmol/L)	0.7	2.1	0.6–2.8
Total Proteins (g/L)	63	70	58–76
Albumin (g/L)	26	23	22–32
Globulin (g/L)	33	38	32–48
Cho (mmol/L)	2.2	2.7	1.4–2.9
Tbil (µmol/L)	0.4	1.5	0.1–3.7
Glucose (mmol/L)	3.9	4.1	3.9–4.7
Calcium (mmol/L)	2.3	2.6	2.2–3.4
Na (mmol/L)	130	134	128–138
K (mmol/L)	3.5	4.7	3.2–5.1
Cl (mmol/L)	98	104	96–106

D1: case donkey 1; D2: case donkey 2; RBC: Red Blood Cells; ALP: Alkaline phosphatase; AST: Aspartate aminotransferase; GGT: Gamma-glutamyl transpeptidase; Creat: Creatinine, BUN: blood urea nitrogen, CK: Creatine phosphokinase, Tri: triglycerides, Tbil: Total bilirubin, Cho: cholesterol, Na: Sodium, K: Potassium, Cl: Chloride. RI = Reference interval. LRL = Low RI. URL = Upper RL [14].

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
