# Peer review of "Management of Thermal Injuries in Donkeys: A Case Report"

_animals, 2020, doi:10.3390/ani10112131_

Round 1

Reviewer 1 Report

As the authors describe, there is very limited information in the literature relating to donkey medicine so it is good to raise awareness of the species. With this in mind however I think it is really important to ensure readers are aware of some of the unique risks associated with donkeys when they are sick. Hyperlipaemia is a secondary complication of any disease process and/or stressful situation so the authors should include some information about this in the introduction. I do have a slight ethical concern about the fact that the donkeys were donated to the university meaning that the clinicians treating them were also their owners and the impact this might have on judgement and decision making so the authors should comment further on this. To ensure the introduction remains concise but contains relevant information the detail about the region could be removed as it adds no value to the content of the manuscript. Reference is made to the Wallace rule of 9 but this is not included in the bibliography.  

As mentioned above the risk of hyperlipaemia would have been high in these donkeys but triglycerides (or failing that cholesterol) was not measured as part of serum biochemistry and the serum was not visually assessed. The authors need to explain why this was not done.

In relation to antibiotic therapy and in light of concerns over resistance the authors need to explain their choice of drugs and dose rate. Typically procaine penicillin needs to be given at 25,000 IU/kg q12hrs and dihydrostreptomycin is not considered useful in equine medicine when compared to gentamicin.

The authors report that when flunixin was discontinued no other NSAIDs were used. Can the authors comment on this as they do report increasing pain in the 2 weeks after flunixin was stopped, which might have been controlled with longer term NSAID use eg. with phenylbutazone.

In the results section the authors state that aloe vera can cause pruritus however this was not deemed to be a problem when it was applied. This contradicts the methods section where pruritus was reported and treated with anti-histamines. How do the authors know that the pruritus was not a result of the aloe vera?

It is incorrect in the conclusion to state that all 4 donkeys suffered from extensive, deep skin burns when D3 and D4 were described as suffering from grade 1 burns over 5% of the body.

Author Response

Response to Reviewer 1 Comments

Point 1 a)As the authors describe, there is very limited information in the literature relating to donkey medicine so it is good to raise awareness of the species. With this in mind however I think it is really important to ensure readers are aware of some of the unique risks associated with donkeys when they are sick. Hyperlipaemia is a secondary complication of any disease process and/or stressful situation so the authors should include some information about this in the introduction.

B)I do have a slight ethical concern about the fact that the donkeys were donated to the university meaning that the clinicians treating them were also their owners and the impact this might have on judgement and decision making so the authors should comment further on this.

 C)To ensure the introduction remains concise but contains relevant information the detail about the region could be removed as it adds no value to the content of the manuscript.

D)Reference is made to the Wallace rule of 9 but this is not included in the bibliography. 

Response 1:

  1. a) The authors agree about the comments of point 1 to mention hyperlipidemia, thank you reviewer.

It was added in lines 75-78

“Donkeys are particularly susceptible to hyperlipidemia in stressful circumstances. It can progress rapidly and is often life threatening. Prompt diagnosis and treatment is required to improve the outcome. If the managed correctly, with an appropriate diet and regular routine preventive healthcare, this risk can be minimized”

  1. b) Dear reviewer, the authors declare that not conflict of interest and also worked always in an ethical manner. The owner of the donkeys was a poor people (with 4 working equids). The day of the fire when the donkey were rescue they mention immediately that can pay the healthcare and clinical management and preferred that we euthanized all. In those moment, we convince them that we can save them, that`s why the owner decided to donate them, and since the next day of the fire we try to found a new place. Hopefully everything was good and finally they were sent to a shelter and live happy and freely.

c)About to delete or remove the region and description of the area of fire, the authors want you considered,  because is special  as mention in the manuscript “Valparaiso has a peculiar topography characterized by 45 hills and the risk for wild fire exists in most parts of rural areas every summer”

  1. d) It was added the reference that reviewer comment in poin 1 .Check line 73

Point 2: As mentioned above the risk of hyperlipaemia would have been high in these donkeys but triglycerides (or failing that cholesterol) was not measured as part of serum biochemistry and the serum was not visually assessed. The authors need to explain why this was not done.

Response 2: The parameter of triglycerides was taken. All the authors we really apologyze for that mistake, also is was add the parameters of Cho. Was add the 2 parameters in the Table 1.

Also we add in the line 132 the TRI and in

133 , the description of the serum.

“It was observed in the clotted blood sample slightly cloudy serum”

In discussion Lines 322-323

All the parameters of complete blood count and serum biochemistry profile were normal since the second analysis

Point 3: In relation to antibiotic therapy and in light of concerns over resistance the authors need to explain their choice of drugs and dose rate. Typically procaine penicillin needs to be given at 25,000 IU/kg q12hrs and dihydrostreptomycin is not considered useful in equine medicine when compared to gentamicin.

Response 3: Dear reviewer we agree with the dose of Procaine penicillin, my mistake, it was used 20.000 IU/kg and about the frequency I used the recommended by Brooke.org (bellow is the link, in the page 109). They recommend BID/SID for this Penicillin. (You can see the change in line 138)

According the Dihydrostreptomycin it is because  in my country is not not sold separately. The bottle contains the both antibiotics and I always want to use Procaine P.

https://www.thebrooke.org/sites/default/files/Professionals/Working%20Equid%20Veterinary%20Manual/WEVM-chapter-5.pdf. Also is in reference 30 of the new version of the manuscript

Point 4: The authors report that when flunixin was discontinued no other NSAIDs were used. Can the authors comment on this as they do report increasing pain in the 2 weeks after flunixin was stopped, which might have been controlled with longer term NSAID use eg. with phenylbutazone.

Response 4: According to the comments in point 4, it is in the manuscript that flunixim was given 10 days to D1 and 12 to D2, then it is also describe that continue con natural product for pain relief (Devil claw, etc.)  

(I added the differences of doses use for flunixim in the donkeys  in line 146-149

“Flunixin was given daily for pain control for 10 days (3 days 1.1 mg/kg q12 h, then 7 days 1.1 mg/kg q 24 h IV) and continue with a natural product (devil's claw 726mg+ Yucca Schidigera extract 954 mg + vitamin B12 10 mcg/ 10mL, Bute-Less Paste, Absorbine, United State), 5 mL per os q 24 h/15 days, supplement to discomfort and pain relief.

And lines 239-241

Flunixin meglumine (1.1 mg/kg) daily for pain was continued for 12 days (3 days 1.1 mg/kg q12 h, then 9 days 1.1 mg/kg q 24 h IV) and the supplement for pain relief same doses rates as D1

Also I added in discussion the next line 360

other NSAIDs could have been used described for donkeys

Point 5 In the results section the authors state that aloe vera can cause pruritus however this was not deemed to be a problem when it was applied. This contradicts the methods section where pruritus was reported and treated with anti-histamines. How do the authors know that the pruritus was not a result of the aloe vera?

Response 5:  I agree with the comments of point 5. It was discuss and added reference.

Review lines 335-339

Aloe vera appeared to be helpful for managing dry skin, which appeared to prevent the skin from sloughing and has allegedly anti-inflammatory effects and promote wound healing [25,26]. It had been reported to cause pruritus in some horses [5], but we cannot mention that aloe vera was the causative agent in D1 and 2. The precise mechanism of post-burn pruritus has not been elucidated in humans and animals, but it appears to have pruritogenic and neuropathic aspects [27].

Point 6: It is incorrect in the conclusion to state that all 4 donkeys suffered from extensive, deep skin burns when D3 and D4 were described as suffering from grade 1 burns over 5% of the body.

Response 6: Thank you reviewer, I agree with the recommendation of point 6. For that reason and for no confuse the lectors I delete  the underlined “All the donkeys in this case series recovered despite extensive and deep skin burns and were retired to an animal shelter after their recovery”.

Review lines 404-405

 All the donkeys in this case series recovered and were sent to an animal shelter after their recovery.

Reviewer 2 Report

Veterinary studies including case reports are lacking for donkeys in general. In addition, there is a lack of information for thermal injuries for all members of the horse family, Equidae, but particularly, the donkey. While this issue may not be seen in the veterinary clinics on a regular basis, it still holds merit in offering knowledge to clinicians that encounter such cases. Nevertheless, while the subject matter holds value to the readers, there are some revisions that are recommended to improve the overall quality of the manuscript.

Introduction:

In line 44, after "animals" and before "was" add "associated with the WUI".
In line 58, add "," after "however" and take out "of" after "involve".
In line 64, change "is" to "are" and add after "Fourth-degree" the word "burns".
At the end of the introduction, line 71, add in a brief statement of what the reader will find in this report (ie. "This report describes the management of veterinary care of thermal injuries in four donkeys").

Materials and Methods:

In line 76, change "is" to "was".
In line 81, add in the temperature of the tap water.
In line 89, change "stabled" to "stable".
In lines 104-105, for the figure description add after "medical attention" "for thermal injuries" and change "soak" to "soaked".
In line 113, take out the "," before "were".
In line 123, replace "," to a "." before which, and then, take out "which" and start the sentence with the following "The first analysis done at day two revealed". Dividing the one sentence, lines 122-126, will make it easier for the reader to follow.
In line 126, take out "In" and start the sentence with "The second analysis", and then after "analysis" add in "at day 8". In the same line remove "in".
In line 131, add "(" before "Pentril".
In line 132-136, for Table 1, it is hard to follow for D1, which analysis is day 2, 8, or fifteen as "*, **, ***" are only under D2. Reformat table to make the different analyses easier to follow.
In line 137, be specific for "following days", was that day fifteen?
In lines 138-139, be specific on the time frame that these positive responses occurred. Which day was this in the treatment of this donkey that these responses became evident?
In lines 139-140, was the edema from days 1-5 starting from the initial presentation? Did it decrease after day 5 or at least remain stable? Remove "her" and replace with "his" since D1 was a male.
In lines 141-142, eye treatment was previously discussed in lines 89-94 so be clear if this is something different from those lines and give specifics on the time frame of the initial treatments for the eyes versus the new treatment.
In lines 142-144, pain treatment is discussed, but it is also discussed in lines 97-101. More than likely, the pain treatment at 142-144 was done after the first few days, but authors need to be specific on the time frame when the initial treatment was ended and the new treatment was started.
In line 145, add "was given" before "for".
In lines 152-153, Figure 3, are both pictures B and C for the same days, ie. 12 days? If both the same day, then change "B" to "B1" and "C" to "B2" and change "D" to "C". Also, if they are the same days, add to the second photo what specific area of the anatomy the reader is looking at as it is hard to discern. If they are different days, correct the days.
In lines 160-161, add "," after "however" and change "were" to "where". Also, add "s" to "blister".
In line 167, add "and" before "nitrofurazone cream".
In lines 168-171, divide into two sentences so that the honey and fly treatment are separate from the nitrofurazone treatment as hard to follow as is.
In line 179, was the damage line specifically on the hoof? Be specific on anatomical location and was it all four legs. Was there any negative responses to thermal injuries associated with the hooves as that should be something noted at this point, and if so, was farrier care specialized for the treatment of these negative responses?
In line 180, change "as" to "with".
In lines 188-189, add a "," before "therefore" and "which".
In lines 191-192, for both of the sentences in these lines add what days that these activities occurred.
In line 194, after "D1" put a ".", and then, start the new sentence by taking out "which" and adding in "The first blood analysis taken at day two revealed" at the beginning of the sentence.
In line 199, change "lavage" to "lavaged" and "clean" to "cleaned".
In lines 201-202, add to both the second and third blood analysis which days they were done.
In line 203, next to "same" add "as D1".
In line 206, for the figure 4 photos, replace photo A with a full body photo of D2 on the right side of the body at day 10 or earlier as it is hard to compare the progression when looking at different sides and at different parts of the body.
In line 211, add change "continue" to "continued" and change "week" to "weeks".
In line 216, add "(D1 and 2) after "cases".
In line 223, change "They" to "D3 and 4".
In line 225, take out "usually" and be specific on what you mean by "easier to manage". Clearly layout any specifics on their management practices for D3 and 4. If you followed the exact management guidelines as a previous case report, then state that you did follow the same guidelines of that specific publication. It is important for readers that may have a donkey patient with first degree burns to see exactly what "easier to manage" means.
In line 228, authors mention blood analyses for D3 and 4, but it would be helpful to see the analyses of these patients, even if "normal", thus, add in another table for D3 and 4.
In line 238, change "oral deworming" to "orally dewormed".
In line 242, indicate in your description which donkey is D1-4, ie. "(left to right is D1, D2, D3, and D4)". Add in at what day this was taken in the treatment process.

Results and Discussion:

In lines 247-250, change "prolong" to "prolonged" and change "euthanize" to "euthanization". Add "," after "treatments" and after "then".
In line 255-256, add "," after "therapy" along with after "stage" and "however".
In lines 257-263, comment on temperature and responses of donkeys with treatments for this case report. Goal is to relate to what was seen in this report.
In line 265, add "," after "however".
In line 267-268, change "manage" to "managed" and add "s" to "complication". Also, within this paragraph, reflect on the use of gauze dressing in this case report and the potential value of this treatment and potential alternative approaches that might apply for donkeys.
In line 282, expand on incidents of eye issues with burns and potential complications. What was seen after donkeys were released and does this relate to previous case reports in equines?
In line 283, is nonsteroidal anti-inflammatories a noted issue with equine burn patients? Expand. Is the prophylactic use of omeprazole a common practice in equine burns? How was this beneficial? Any additional recommendations? Expand.
In lines 287-290, is the diet in this case report complementary of other burn case reports? Is it typical for donkeys? Discuss potential issues with thermal injuries and diets and additional considerations with the low BCS.
In lines 291-293, is fans or AC stalls recommended for thermal injuries? Compare to other case reports.
In lines 294-296, add in number of veterinarians and other staff that assisted with these cases and expand on this discussion of teamwork giving the reader some idea of the extent of staffing needed to monitor and treat these patients.

Conclusions:

In line 299, change "this" to "these".
In line 301, it mentions "retired", which gives the impression that the animals required additional specialized management and was not serviceable. Authors should conclude with a statement as to the extent of how serviceable these animals were after treatment and specialized management needed after being released so that clinicians and owners know the advantages and disadvantages to treatment versus euthanization.

Author Response

Response to Reviewer 2 Comments

Introduction:

Point 1: In line 44, after "animals" and before "was" add "associated with the WUI".

Response 1: It was added in the paragraph according your recommendation 1. The change is in line. 45

Point 2: In line 58, add "," after "however" and take out "of" after "involve".

Response 2: The changes were made in the paragraph according your recommendation 2, the change  is in line. 61

Point 3: In line 64, change "is" to "are" and add after "Fourth-degree" the word "burns".

Response 3: The changes were made in the paragraph according your recommendation 3, the change  is in line. 67

Point 4: At the end of the introduction, line 71, add in a brief statement of what the reader will find in this report (ie. "This report describes the management of veterinary care of thermal injuries in four donkeys").

Response 4: Thanks for your comment. It was add in the last paragraph (line 79-80) This report describes the medical management of veterinary care of extensive thermal injuries in four donkeys

Materials and Methods:

Point 5: In line 76, change "is" to "was".

Response 5: It was change “as”   o  “ was”. Thanks. The change is in line 85

Point 6: In line 81, add in the temperature of the tap water.

Response 6: The suggested temperature of point 6 was added. The change is in line 90

Point 7: In line 89, change "stabled" to "stable".

Response 7:Thanks reviewer, the change was made. The change  is in line 98

Point 8: In lines 104-105, for the figure description add after "medical attention" "for thermal injuries" and change "soak" to "soaked".

Response 8: The suggestion of point 8 were made. The change is in lines 111-112

Point 9: In line 113, take out the "," before "were".

Response 9: It was remove the “,”, response for point 9. The change is in line 120

Point 10: In line 123, replace "," to a "." before which, and then, take out "which" and start the sentence with the following "The first analysis done at day two revealed". Dividing the one sentence, lines 122-126, will make it easier for the reader to follow.

Response 10: Thanks for the recommendation, the suggested changes were made. Review change in line. 129-134

Point 11: In line 126, take out "In" and start the sentence with "The second analysis", and then after "analysis" add in "at day 8". In the same line remove "in".

Response 11: The suggested changes were made in the paragraph indicated. The change is in line 135

Point 12: In line 131, add "(" before "Pentril")

Response 12: thanks for the recommendation I added “(“. The change is in line 139

Point 13: In line 132-136, for Table 1, it is hard to follow for D1, which analysis is day 2, 8, or fifteen as "*, **, ***" are only under D2. Reformat table to make the different analyses easier to follow.

Response 13: Thanks for your comments of point 13 reviewer 1. We change the format and we believe that will clarify all, Also delete **, etc etc. we move the table 1 from original lines, now is in lines 165-169

Point 14: In line 137, be specific for "following days", was that day fifteen?

Response 14: I agree about the comment of point 14, it was added the days. The change is in line 141

Point 15: In lines 138-139, be specific on the time frame that these positive responses occurred. Which day was this in the treatment of this donkey that these responses became evident?

Response 15: I agree about the comment of point 15, it was added the response in the paragraph . The change is in line 143-144

Point 16: In lines 139-140, was the edema from days 1-5 starting from the initial presentation? Did it decrease after day 5 or at least remain stable? Remove "her" and replace with "his" since D1 was a male.

Response 16: I agree about the comment of point 16, it was added the responses in the paragraphs . The changes is in lines 144-145

Point 17: In lines 141-142, eye treatment was previously discussed in lines 89-94 so be clear if this is something different from those lines and give specifics on the time frame of the initial treatments for the eyes versus the new treatment.

Response 17: We agree with the comments of point 17, thanks reviewer. There not separately treatments, is the same, then we clarify. We move the sentence from lines 141-142 in the original manuscript. Now all the sentence are in lines 99-105.

Treatment with ophthalmic antibiotic ointment containing zinc bacitracin 400 iu/g, neomycin sulphate 3.5 mg/g and polymyxin B sulphate 6,000 IU/g q 12 h (Oftabiotico, Saval S.A. Laboratory, Santiago, Chile), ophthalmic atropine sulfate drops q 12 h x 2 days (10mg /ml, Atropina 1%, Saval S.A. Laboratory, Santiago, Chile) and dextran-70 lubricant drops 1 mg with hidroxipropilmetilcelulosa 3 mg (Tears Naturale II, Alcon Laboratory Chile Ltda, Santiago, Chile) several times a day was initiated. The eyes were treated with triple antibiotic ointment for six weeks, and the natural tears drops for four weeks

Point 18: In lines 142-144, pain treatment is discussed, but it is also discussed in lines 97-101. More than likely, the pain treatment at 142-144 was done after the first few days, but authors need to be specific on the time frame when the initial treatment was ended and the new treatment was started.

Response 18: We agree reviewer with the comments of point 18, it is all clarify in the lines:

We found that we repeat management in the original manuscript 97-101 and lines 154-156, now the correct is lines 178-180, for that reason we delete the lines 97-101, also clarify the point 18, about the days

Similar comment made the reviewer 1, point 4

The changes are in lines 178-180

Daily skin care initially required sedation xylazine 0.5 mg/kg IV, with butorphanol tartrate 0.01 mg/kg IV (10 mg/ mL, Torbugesic, Zoetis, NJ, United State) as potent analgesic for the first six days in the clinic. On day seven, sedation and potent analgesic for treatments was no longer required

Point 19: In line 145, add "was given" before "for".

Response 19: The change was made, thanks reviewer. Response for point 19. The change is in line 150

Point 20: In lines 152-153, Figure 3, are both pictures B and C for the same days, ie. 12 days? If both the same day, then change "B" to "B1" and "C" to "B2" and change "D" to "C". Also, if they are the same days, add to the second photo what specific area of the anatomy the reader is looking at as it is hard to discern. If they are different days, correct the days.

Response 20: we agree with your comments of point 20. The changes wer made in line 176 and also in the photos

Point 21: In lines 160-161, add "," after "however" and change "were" to "where". Also, add "s" to "blister".

Response 21: The change was made, thanks reviewer for the recommendation. The change is in line 184-185

Point 22: In line 167, add "and" before "nitrofurazone cream".

Response 22: The change was made, now the sentences is in line. The change is in line 191

Point 23: In lines 168-171, divide into two sentences so that the honey and fly treatment are separate from the nitrofurazone treatment as hard to follow as is.

Response 23: The change that you suggest were made, it was divided in 2 sentences (the wound treatment and apart the fly repellent. The change is in lines 192-196

Nitrofurazone was used in areas with exudation and erythema and locally raw honey on areas with less exudation and erythema. Dry skin areas were treated with jelonet then changed to Buddleja globosa cream. Fly repellent was used around wounds (dichlorvos 1250 mg +triclosan, 500 mg/ 100 g, Moskation larvicida spray, Drag Pharma Laboratory Chile Invetec S.A., Santiago, Chile).

Point 24: In line 179, was the damage line specifically on the hoof? Be specific on anatomical location and was it all four legs. Was there any negative responses to thermal injuries associated with the hooves as that should be something noted at this point, and if so, was farrier care specialized for the treatment of these negative responses?

Response 24:we agree with the point 24, we added the necessary description according the hoof. (Similar comments made the reviewer 3, point 7) You can check the

D1

Line:   87-88  All serous exudate was observed over the coronary bands in front limbs

Line s 242-244

His coronary bands from the front limbs did not separate significantly, but a horizontal line of damage was evident around the hoof wall (1 cm below the coronary band), walk and trot sound.

D2 LINES 203.205

The left coronary band lesion increasing in size and lameness was graded 2/5. Radiographs were taken, which revealed no sign of distal phalangeal rotation

D2 LINES 241-244

 After six weeks she could canter when turn out into a paddock in the evenings or early mornings when the outdoors temperatures were cooler. His left coronary band from the hind limb did not separate, but a horizontal line of damage was evident around the hoof wall (1 cm below the coronary band)

Point 25: In line 180, change "as" to "with".

Response 25: The recommendation of Point 25 was made.  The change is in line 206

Point 26: In lines 188-189, add a "," before "therefore" and "which".

Response 26: The suggestion of point 26 was added. The change is in line 214

Point 27: In lines 191-192, for both of the sentences in these lines add what days that these activities occurred.

Response 27: we agree about the comments of point 27, we added the information required. The changes are in lines 217-218

“Fluids were discontinued on the second day at the clinic, when the laboratory report indicated adequate hydration, also she was eating hay with a good appetite at this time”.

Point 28: In line 194, after "D1" put a ".", and then, start the new sentence by taking out "which" and adding in "The first blood analysis taken at day two revealed" at the beginning of the sentence.

Response 28: The changes suggested in Point 28 were made. The change is in lines 220

Point 29: In line 199, change "lavage" to "lavaged" and "clean" to "cleaned".

Response 29: The suggestion of point 29 to change the words to past tense was made. The change is in line 227

Point 30: In lines 201-202, add to both the second and third blood analysis which days they were done.

Response 30: The suggestion of point 30 were made. The change is in lines 230-232

Point 31: In line 203, next to "same" add "as D1".

Response 31: The suggestion of point 31 was added. The change is in line 233

Point 32: In line 206, for the figure 4 photos, replace photo A with a full body photo of D2 on the right side of the body at day 10 or earlier as it is hard to compare the progression when looking at different sides and at different parts of the body.

Response 32: Dear reviewer 2, according the pointe 32, unfortunately we do not have more pictures of the other side to added another photo of the right side. We understand your point of view, please let as now if we remove the photo A and only shows the B ?

Point 33: In line 211, add change "continue" to "continued" and change "week" to "weeks".

Response 33: The suggestion of point 33, of change the past tense of continue was did it, also add an s to word week. The changes is in line 241

Point 34: In line 216, add "(D1 and 2) after "cases".

Response 34: The change suggested in point 34 was did it, thank you reviewer. The change is in line 249

Point 35: In line 223, change "They" to "D3 and 4".

Response 35: The suggestion of point 35, was did it. The change is in line 255

Point 36: In line 225, take out "usually" and be specific on what you mean by "easier to manage". Clearly layout any specifics on their management practices for D3 and 4. If you followed the exact management guidelines as a previous case report, then state that you did follow the same guidelines of that specific publication. It is important for readers that may have a donkey patient with first degree burns to see exactly what "easier to manage" means.

Response 36: Accordind with the comments of point 36, it was clarify the sentence. The change is line 257-261

Superficial burns healed without complication. The areas affected became dry between 5-8 days. The wounds from the skin was periodically lavaged and cleaned using disinfectant and similar topic cream described in D1. It was necessary to used xilazyine and buthorphanol tartrate previously to wound management for 2 days since they arrived to the clinic. After 3 weeks most of the burned areas had healed

Point 37: In line 228, authors mention blood analyses for D3 and 4, but it would be helpful to see the analyses of these patients, even if "normal", thus, add in another table for D3 and 4.

Response 37: I am agree with the recommendation of point 37, it was add a new table 2 for D3-D4. The change is in line 282-286

Point 38: In line 238, change "oral deworming" to "orally dewormed".

Response 38: I agree and have made the suggested change of point 38. The change is in line 274

Point 39: In line 242, indicate in your description which donkey is D1-4, ie. "(left to right is D1, D2, D3, and D4)". Add in at what day this was taken in the treatment process.

Response 39: According the comments of point 39, we added the information required. Change is in line 282

Results and Discussion:

Point 40: In lines 247-250, change "prolong" to "prolonged" and change "euthanize" to "euthanization". Add "," after "treatments" and after "then".

Response 40: Thanks reviewer, I agree and have made the suggested change of point 40. The change is in lines 293-295

Point 41: In line 255-256, add "," after "therapy" along with after "stage" and "however".

Response 41: The add of “,” of point 41 it was made it. The change is in lines 306-307

Point 42: In lines 257-263, comment on temperature and responses of donkeys with treatments for this case report. Goal is to relate to what was seen in this report.

Response 42: it was added a sentence according the point 42, Now is in the line 312-313

Point 43: In line 265, add "," after "however".

Response 43:  it was add the suggestion of point 43. The change is in lines 325

Point 44: In line 267-268, change "manage" to "managed" and add "s" to "complication". Also, within this paragraph, reflect on the use of gauze dressing in this case report and the potential value of this treatment and potential alternative approaches that might apply for donkeys.

Response 44: The changes suggested in point 44 was made, also added more. The change is in lines 330-333

The new paragraph is “

There are several methods to treat bum wounds in the horse, and the choice depends on the extent and location of the injury. Full-thickness burns can be managed by occlusive dressings (closed technique), continuous wet dressings (semiopen technique), eschar formation (exposed technique), or excision and grafting [4]. In this case report bandaging of the burned limbs was not performed as mentioned in burned horses [5], and was managed as open wounds with no complications. The use of the compression garments or compression suits post-burn injury is described as an important component of human patient’s rehabilitation program [5,22,23].

Point 45: In line 282, expand on incidents of eye issues with burns and potential complications. What was seen after donkeys were released and does this relate to previous case reports in equines?

Response 45: We agree the comments of point 45. We added literature and more discussion as you recommend. The changes is in lines 351-358

The donkeys with corneal ulcers (D1 and 2), responded well to treatment and healed with no complications. Human studies report 7.7%−18% the incidence of chemical and thermal injuries to the eye of all ocular trauma, with corneal ulcers and blefarospasm [29]. Burned horse may have blepharospasm, epiphora, or both, which signify corneal damage [4]. Ocular emergency treatment by thermal injuries in humans it is described [29]. Case reports in horses injured in open range fires that, reported corneal ulcers were, successfully treated with twice daily treatments with topical antibiotics (zinc bacitracin, neomycin sulphate and polymyxin B sulphate), plus daily cloxacillin benzathine eye ointment [5].

Point 46: In line 283, is nonsteroidal anti-inflammatories a noted issue with equine burn patients? Expand. Is the prophylactic use of omeprazole a common practice in equine burns? How was this beneficial? Any additional recommendations? Expand.

Response 46:  The use of omeprazole is only prevention of EGUS when you use several drugs and many days of NSAIDs, but is not specific prophylactic in equine burns. Similar comments made reviewer 3. The true in this report, Omeprazole tablet it was use by the economic possibility of that moment. We have not proof  that it was beneficial. We decide to write this in the lines  361-366

The prophylactic use of omeprazole may have been beneficial in preventing gastric ulceration with long-term use of NSAIDs, in this report it was use omeprazole (human tablets) by the economic possibility of that moment, but do not have evidence that was beneficial in the donkeys, the authors recommend approved omeprazole product manufactured to horses, using the standard equine dosage, described appropriate for donkeys [14].

Point 47: In lines 287-290, is the diet in this case report complementary of other burn case reports? Is it typical for donkeys? Discuss potential issues with thermal injuries and diets and additional considerations with the low BCS.

Response 47: According with suggestion of point 47 we added the next lines 387-395

A hallmark of burn injury is a hypermetabolic response that results in significant pathological alterations in a number of tissues [35]. Studies in large animal models of burn hypermetabolism showed that 25% TBSA burn can generate a hypermetabolic response greater than smaller animals and closer to that seen in human patients [36,37]. It is recommended gradually increasing the grain, adding fat in the form of vegetable oil, and offering free-choice alfalfa hay increase caloric intake as nutritional needs for burned horses [4]. An anabolic steroid may be used to help restore a positive nitrogen balance [4]. The primary goal of this response is to provide adequate energy levels to maintain organ function and whole-body homeostasis. In addition to the food offered, multivitamin and mineral supplements and vegetable oil to increase the energy intake were added.

Point 48: In lines 291-293, is fans or AC stalls recommended for thermal injuries? Compare to other case reports.

Response 48: We agree with the comment of you made in point 48, we change all the structure of the paragraph. Also the paragraph was move to lines 339-345

The occasional pruritis presented in D1 and 2 were controlled by the use of chlorphenamine. Many burned equine patients are pruritic, and measures must be taken to prevent self-mutilation of the wound [4].Reserpine had been used in burned horses for this complication, decreasing the urge to scratch by successfully breaking the itch-scratch cycle [4,5], also there is mentioned the use of air conditioning fans directed to the stall which can helped [5]. The same author mentioned that topical corticosteroid creams were ineffective in controlling the itching in horses after burned [5].

Point 49: In lines 294-296, add in number of veterinarians and other staff that assisted with these cases and expand on this discussion of teamwork giving the reader some idea of the extent of staffing needed to monitor and treat these patients.

Response 49: We added the suggestion of point 49. The changes is in lines 396-399

In veterinary emergency medicine and critical care, teamwork to provide adequate patient care is essential. The entire team consisted of two veterinarians, a technician and 8 graduate students on a permanent basis day and night. The care and devotion provided to each animal by the veterinarian staff, highlights the concept of animal-human bond and the dedication to their patients.

Conclusions:

Point 50: In line 299, change "this" to "these".

Response 50: I agree the recommendation of point 50. The change is in line 403

Point 51: In line 301, it mentions "retired", which gives the impression that the animals required additional specialized management and was not serviceable. Authors should conclude with a statement as to the extent of how serviceable these animals were after treatment and specialized management needed after being released so that clinicians and owners know the advantages and disadvantages to treatment versus euthanization.

Response 51:Dear reviewer about the point 51, we understand that was misunderstood with the word “retired”, we change the word to “sent”.

Now the sentence is in line 404-405

“All the donkeys in this case series recovered and were sent to an animal shelter after their recovery.

Currently They live freely in the shelter.

Reviewer 3 Report

Dear authors,

It is true that there little literature on the (treatment of) burn injuries in large animals/equids and what is available often relates to barn fires, which differ in a number of respects from wild fires. Sharing experiences via case reports such as this one can help others when treating similar cases. Your experience in dealing with these donkeys (only 2 of which were severely affected) could be useful in that respect.

I feel it would be beneficial for a native English speaker to correct the manuscript, which would improve clarity.

Also, some more detail/clarification and discussion/justification of choices made should be included.

A number of specific comments are included below:

12: Simple summary: Reports or descriptions of medical management of thermal injuries.....

44: domestic animals

45: the numbers don't had up (to 177)

The introduction mentions that thermal damage (through smoke inhalation) can affect the airways. Please provide some information here about that aspect of thermal injuries, possibly including the fact that this is less common in wild fires than in barn fires.

80: how long was the donkey hosed?

87: do you know why he wasn't eating well? Was there also facial swelling/edema?

How did the damage/lesions of the coronary bands progress? Were there any cases of laminitis? Please elaborate.

There is debate/discussion regarding the use of antimicrobials in (equine) burns victims. Please discuss.

Table 1: please indicate the day of sampling in the table itself (and not with asterisks), increase spacing between columns and/or add gridlines for clarity (for example it is not entirely clear what the CK values of the 2 donkeys was on day 2) and was the albumin of the second donkey really 1.9 on day 8??

145: Is there any evidence to suggest that omeprazole tablets (rather than buffered paste or enteric coated) are effective in donkeys (or equids)?

159: what was the initial interval of lavage/debridement and did that change during the course of treatment. Every 96 hours would seem too long in the early stages of treatment. Please comment

178: how well had the lesions healed, was there hair growth? Complete healing (with hair growth) would suggest the burns had not actually been 3rd degree.

179: how did coronary band damage (separation) progress?

Figure 4(B) suggests that the TBSA in donkey 2 was (considerably) more than 30%. In my experience, estimating the severity and extent of burn injury just after being sustained is very difficult. Please comment.

Figure 6: please indicate how long after the fire this picture was taken. Close-up pictures of donkeys 1 and 2 would be useful to help assess the quality of healing.

270: you indicate debridement/wound management was well tolerated but initially the donkeys required sedation. Please comment/clarify.

285: (again) I question the efficacy of pure omeprazole

I think there could be more discussion around the treatments provided, with comparisons with what has been reported for horses and/or humans.

Author Response

Response to Reviewer 3 Comments

Point 1: 12: Simple summary: Reports or descriptions of medical management of thermal injuries.....

Response 1: Thank you reviewer, the suggestion of point 1 was made. The change is in line 12

Point 2: 44:domestic animals

Response 2: The suggested word in point 2 was added. The change is in line 44

Point 3: 45: the numbers don't had up (to 177)

Response 3: Thanks, was a mistake, the number is 146, was changed. The change suggested in point 3 is in line 45

Point 4: The introduction mentions that thermal damage (through smoke inhalation) can affect the airways. Please provide some information here about that aspect of thermal injuries, possibly including the fact that this is less common in wild fires than in barn fires.

Response 4:Thanks reviewer by your comments of point 4. (in red), it was added information in the introduction , lines 50-54

During fires, thermal injuries are caused by direct exposition to the flames and/or inhalation of toxic gases [7]. Inhalation injury is a common sequel of closed-space fires and develops through three mechanisms: direct thermal injury, carbon monoxide poisoning, and chemical insult [4]. Horses exposed can suffer respiratory injury of varying degrees, ranging from mild irritation to severe smoke inhalation-induced airway or lung damage [8].

And also in discussion modification, lines 317-321

D1 showed respiratory signs from smoke inhalation, which resolved in 3 days with the established treatment. The area of donkeys where donkeys were rescued from fire, that day was windy, which helped that sweep away smoke and ash in the atmosphere, that could be the reason that D2 to 4 not presented problems in the respiratory system. Case report of horses injured in open range fires described one horse with problems by smoke inhalation which resolved in 24 h [5].

Point 5 80: how long was the donkey hosed?

Response 5: The suggested time in point 5 was added. The information is in  line 91

Point 6: 87: do you know why he wasn't eating well? Was there also facial swelling/edema?

Response 6: Thanks reviewer, I agree with your question of point 6 and I added the next in the text (changes in lines 95-96

There was also facial, abdominal and limb edema. He was eating and drinking with difficulty, dysphagia possibly caused by edema and pain.

Point 7: How did the damage/lesions of the coronary bands progress? Were there any cases of laminitis? Please elaborate.

Response 7:Dear reviewer similar comment made reviewer 2, point 24. I clarify more in lines

D1

Line:   87-88  All serous exudate was observed over the coronary bands in front limbs

Line s 242-244

His coronary bands from the front limbs did not separate significantly, but a horizontal line of damage was evident around the hoof wall (1 cm below the coronary band), walk and trot sound.

D2 LINES 203.205

The left coronary band lesion increasing in size and lameness was graded 2/5. Radiographs were taken, which revealed no sign of distal phalangeal rotation

D2 LINES 241-244

 After six weeks she could canter when turn out into a paddock in the evenings or early mornings when the outdoors temperatures were cooler. His left coronary band from the hind limb did not separate, but a horizontal line of damage was evident around the hoof wall (1 cm below the coronary band)

Point 8: There is debate/discussion regarding the use of antimicrobials in (equine) burns victims. Please discuss

Response 8: we agree reviewer, thanks for the suggestion in point 8. It was the following lines 374-386

Preventive antibiotic therapy was used in D1 and 2, but the literature mentioned that role of prophylactic antibiotics for severe burns is controversial, both in humans and horses [33]. Therefore their use has not been advocated in recent guidelines or recommendations owing to a lack of evidence for efficacy and induction of antibiotic resistance. Systemic antibiotics do not favorably influence wound healing, fever, or mortality and can encourage the emergence of resistant microorganisms. Additionally, circulation to the burned areas is often compromised, making it highly unlikely that parenteral administration of antibiotics can achieve therapeutic levels to the wound [4]. However, topical antimicrobials have been the mainstay of nonsurgical burn treatment [16]. In human reports the use of prophylactic antibiotics may result in improved 28-day in-hospital mortality in mechanically ventilated patients with severe burns but not in those who do not receive mechanical ventilation [34]. The use of antibiotics had been recommended in equine burned cases because of puncture wounds that occurred while attempting to escape the fire [5]. We do not find evidence that systemic therapy was useful, and we will not recommend it in future cases.

Point 9: Table 1: please indicate the day of sampling in the table itself (and not with asterisks), increase spacing between columns and/or add gridlines for clarity (for example it is not entirely clear what the CK values of the 2 donkeys was on day 2) and was the albumin of the second donkey really 1.9 on day 8??

Response 9: Please provide your response for Point 9. 165-169

Point 10 .145: Is there any evidence to suggest that omeprazole tablets (rather than buffered paste or enteric coated) are effective in donkeys (or equids)?

Response 10:I agree with you in the point 10. The true in this report, Omeprazole tablet it was use by the economic possibility of that moment. We have not proof  that it was beneficial. The same comments made reviewer 2, We decide to write this in the lines 361-366

The prophylactic use of omeprazole may have been beneficial in preventing gastric ulceration with long-term use of NSAIDs, in this report it was use omeprazole (human tablets) by the economic possibility of that moment, but do not have evidence that was beneficial in the donkeys, the authors recommend approved omeprazole product manufactured to horses, using the standard equine dosage, described appropriate for donkeys [14].

Point 11: 159: what was the initial interval of lavage/debridement and did that change during the course of treatment. Every 96 hours would seem too long in the early stages of treatment. Please comment

Response 11: Dear reviewer we agree with your comments of point 11,  we write wrong 96 , it was between 24-36 h. The change is in lines

Point 12 : 178: how well had the lesions healed, was there hair growth? Complete healing (with hair growth) would suggest the burns had not actually been 3rd degree.

Response 12:We clarify the comments  of point 12, about the hair growth. Thank you reviewer, but we suggest that was 3 degree

Clarify in lines:93-95

Approximately 12 h after the incident, the estimated TBSA was approximately 40–50%. Most of the affected areas were classified as deep second-degree burns with some small areas of third-degree burns on the distal area of front limbs and face

Lines 201-202

After 3 months, all treatments were discontinued, and most of the burned areas had healed, but the area of face and front limbs without hair growth

Point 13: 179: how did coronary band damage (separation) progress?

Response 13:  It was added information about the comments of point 13, in lines 370-373

Coronary band effusion was noted on the first day on D1 and 2; never showed separation, but resulted in abnormal hoof growth until the old hoof was replaced, observed in exam of the donkeys 8 month after the fire. The thickened hoof wall distal to the coronary band protected the majority of the laminae from thermal injury [5].

Point 14. Figure 4(B) suggests that the TBSA in donkey 2 was (considerably) more than 30%. In my experience, estimating the severity and extent of burn injury just after being sustained is very difficult. Please comment

Response 14: Thanks for he detail it is in the introduction The “Wallace Rule of Nines” system used in human medicine, estimates the prognosis according to the extent of the burn. This system divides the body surface area into regions that represent multiples of nine, allocates body regions as follows: each limb, head, neck, thorax and abdomen represent 9% and when calculated again the area of TBSA IN D2 I MENTION.    Her burns were classified as second-degree with some areas with third-degree (perineal area

THEN  calculating was at least 45%, AND CHANGE THE SENTENCE TO 40-50%, NO 30%

Review line 211-212

We added some sentence by your suggestion of evaluation of burn injury in DISCUSSION  In lines 300-306

Furthermore, estimates of burn size and depth made before admission to a burn centre and even by experts have consistently been shown to be inaccurate in spite of standardization attempts and the availability of tools such as the Rule of Nines, or other methods described [9,16]. Given the unreliability of burn size and depth assessment by clinicians who are not burn experts, development a scale for burn patients in equids is essential. New approaches development for human medicine include the use of computer-assisted programs to improve burn size estimation [17,18]. The treatment of these cases with extensive burns was the major challenge

Point 15: Figure 6: please indicate how long after the fire this picture was taken. Close-up pictures of donkeys 1 and 2 would be useful to help assess the quality of healing.

Response 15:The observation in point 15, was made. The picture now it is with little more close up, but not to much, because this picture if we close up more, the quality will really decrease. Sadly I don `t have the original, obviously it is with the quality required for the animals journal. If you consider is not good , finaly we can remove the picture.

Also and write the day that was picture was taken since they arrive

Review lines 280-282

Point 16: 270: you indicate debridement/wound management was well tolerated but initially the donkeys required sedation. Please comment/clarify.

Response 16: According the comments of point 16, it was clarify the sentence, I remove some words. Now is in line 334

Cleaning with mild soap and fluids were used to facilitate softening and removal of dead skin.

Point 17. 285: (again) I question the efficacy of pure omeprazole

Response 17: I agree with you, have not proof. As I mention before response 10.

 The true in this report, Omeprazole tablet it was use by the economic possibility of that moment. We have not proof  that it was beneficial. The same comments made reviewer 2, We decide to write this in the lines 361-366

The prophylactic use of omeprazole may have been beneficial in preventing gastric ulceration with long-term use of NSAIDs, in this report it was use omeprazole (human tablets) by the economic possibility of that moment, but do not have evidence that was beneficial in the donkeys, the authors recommend approved omeprazole product manufactured to horses, using the standard equine dosage, described appropriate for donkeys [14].

Point 18: I think there could be more discussion around the treatments provided, with comparisons with what has been reported for horses and/or humans.

Response 18: Dear reviewer, thanks for the comments of point 18. We added more discussion of the case reports. The new discussion:   lines , 289-404

Round 2

Reviewer 1 Report

The authors have responded positively to the reviewers' comments and have made significant alterations to the manuscript providing additional information as required. I have the following comments:

As previously mentioned and agreed by the authors 2/4 donkey shave extensive, severe burns and 2/4 had limited, mild burns so in L13 and L47 the authors need to remove the words "extensive" and "severe", respectively to reflect the range of presentations.

In paragraph L387-395 the authors discuss the increased nutritional requirements for burn cases and reference options available to increase calorie content in a horse's diet. There should be some reference made to the different physiology of the donkey's GI system meaning that care needs to be taken introducing grain to the diet as this can be a risk factor for gastric ulceration and hyperlipaemia (there are papers available for reference). Introducing oil in palatable amounts and high energy fibre products are preferable options. Mention should also be given in this paragraph to monitoring triglyceride levels during treatment and recovery in view of the risk of developing a negative energy balance if calorie intake is not sufficient.

The authors have provided more detail around the use of NSAIDs and that others could have been considered once flunixin was stopped and I accept that for D1 for 15d after flunixin was stopped there was no obvious pain however when pain was identified in the following 2 week period the authors should explain why they did not review pain management at this stage and consider the use of alternative therapies to provide better control. 

Author Response

Response to Reviewer 1 Comments

Point 1 : As previously mentioned and agreed by the authors 2/4 donkey shave extensive, severe burns and 2/4 had limited, mild burns so in L13 and L47 the authors need to remove the words "extensive" and "severe", respectively to reflect the range of presentations.

Response 1: Dear reviewer, according the recommendation of point 1, the changes are in lines 13. And the comment of L47 (same as reviewer 3, point 1). We decide to change the sentence, review line 47, 48.

POINT 2: In paragraph L387-395 the authors discuss the increased nutritional requirements for burn cases and reference options available to increase calorie content in a horse's diet. There should be some reference made to the different physiology of the donkey's GI system meaning that care needs to be taken introducing grain to the diet as this can be a risk factor for gastric ulceration and hyperlipaemia (there are papers available for reference). Introducing oil in palatable amounts and high energy fibre products are preferable options. Mention should also be given in this paragraph to monitoring triglyceride levels during treatment and recovery in view of the risk of developing a negative energy balance if calorie intake is not sufficient.

Response 2: Dear reviewer according the comment of point 2, we added more information.

Lines 388-404. But modification is the is the underlined paragraph

A hallmark of burn injury is a hypermetabolic response that results in significant pathological alterations in a number of tissues [36]. Studies in large animal models of burn hypermetabolism showed that 25% TBSA burn can generate a hypermetabolic response greater than smaller animals and closer to that seen in human patients [37,38]. It is recommended to gradually increase the amount of grain, adding fat in the form of vegetable oil, and offering free-choice alfalfa hay increase caloric intake as nutritional needs for burned horses [4]. An anabolic steroid may be used to help restore a positive nitrogen balance [4]. Donkeys are highly efficient at digesting poor nutritional quality fiber and have lower energy requirements than horses and ponies of similar size and even lower in sick donkeys [14,39,40]. Equine feedstuff based upon cereals or containing high levels of molasses should be avoided. They have been shown to be risk factors for the development of gastric ulcers, laminitis, hyperlipidemia and fatty liver disease in donkeys [14,40]. Early intervention to restore a positive energy balance even before triglyceride values are known greatly increase the chance of survival. The restoration of a positive energy balance will stimulate endogenous insulin secretion and switch off lipolysis. Soya bean meal or alfalfa are excellent sources of digestible protein for convalescent donkeys [14,40].The primary goal of this response is to provide adequate energy levels to maintain organ function and whole-body homeostasis. In addition to the food offered, multivitamin and mineral supplements and vegetable oil to increase the energy intake were added.

Also it was added the reference 39 and 40  

Point 3: The authors have provided more detail around the use of NSAIDs and that others could have been considered once flunixin was stopped and I accept that for D1 for 15d after flunixin was stopped there was no obvious pain however when pain was identified in the following 2 week period the authors should explain why they did not review pain management at this stage and consider the use of alternative therapies to provide better control.

Response 3: Dear reviewer according with the comments of point 3, our explanations: the donkey’s pain described was controlled with the products that we mention in the manuscript.

All the cases, specially D1-D2 tolerated the clinical managements after we finish with NSAIDs- We always feed them to relax while the clinical management was made,. That was the reason we decide not use more NSAIDs

In the reference 5, they described the use in burned horse’s pain control between 2-40. Days, depend of the case.

We made several controls per day and weeks with the team. Always control the animal and we decide not use after we described.

But we decide to clarify your doubt in the line 141-142:

During the hospitalization period (90 days), the burned surfaces, appetite and pain level of the donkey were monitored

With this sentence clarify that all the period in the clinic the donkeys were monitored.

Reviewer 3 Report

Dear authors,

This case series, describing the treatment and outcome of several donkeys suffering burn injuries, may be useful for other veterinarians treating similar cases, providing guidelines for treatment and an indication of what they may expect in terms of timeline and prognosis. 

I appreciate English is probably not your first language, but feel the manuscript would be very much improved by extensive English language editing. This will improve clarity in places and readability overall.

In addition, I have a number of specific comments, suggestions and questions:

47: four donkeys suffered burns, two of them severe, requiring....

54: extent of the burns can vary...

69: (TBSA) affected

71: maybe you could comment on the suitability of the Wallace rules of nine system for estimating TBSA in horses

77: this sentence currently suggests that hyperlipidemia can be managed by preventive healthcare. That is not the case. Please clarify.

79: I think you could/should indicate that 2 of the 4 donkeys were (much) less severely affected.

88: range of motion of which joint(s)? The coronary bands don't have a ROM

99: what were the signs associated with uveitis?

121: was endoscopy of the trachea performed? If so, what was seen?

321: a (very) brief comparison with barn fires could be made here.

333: and reported for horses (reference 5)

144: where (on the body) was the edema present?

204: Presumably the horizontal line moved further down the hoof over time. Please clarify (as is done briefly in the discussion)

Author Response

Response to Reviewer 3 Comments

Point 1: 47: four donkeys suffered burns, two of them severe, requiring....

Response 1: It was made the change of point 1. Review line 47-48

Point 2: 54: extent of the burns can vary...

Response 2: The change suggest of point 2, is in line 54

Point 3: 69: (TBSA) affected

Response 3: The change suggest of point 2, is in line 69

Point 4: 71: maybe you could comment on the suitability of the Wallace rules of nine system for estimating TBSA in horses

Response 4: Dear reviewer, about comments of point 4. We suggest check the lines 302-307

Furthermore, estimates of extend of burn and depth made before admission to a burn Centre and experts in this field, have consistently shown to be inaccurate despite standardization attempts and available of tools such as the Rule of Nines [9,16]. Given the unreliability of burn size and depth assessment by clinicians who are not burn experts, development a scale for burn equids patients we believe is essential. New approach from human medicine include the use of computer-assisted programs to improve burn size estimation [17,18]

Point 5: 77: this sentence currently suggests that hyperlipidemia can be managed by preventive healthcare. That is not the case. Please clarify.

Response 5: Dear reviewer, about the comment of point 4, it was a paragraph from the reference 14 (page 87) of the manuscript, and I agree with the book, because it is taking about hyperlipaemia and it is correct if as the sentence mention:  Prompt diagnosis….and also in an stressfull situation (like from the manuscript), the donkey managed correctly, with an appropriate diet and regular routive preventive healthcare , the risk of suffer hyperlipaemia  can be  minimized

Please read againd the lines 75-78

Point 6: 79: I think you could/should indicate that 2 of the 4 donkeys were (much) less severely affected.

Response 6: According of the comment of point 6, we remove the word “extensive”, and clarify all. The description of every case it is mention.

Check Line 79 and 80

“This report describes the medical management of veterinary care of thermal injuries in four donkeys”

Point 7: 88: range of motion of which joint(s)? The coronary bands don't have a ROM

Response 7: According of comment of point 7, we talk about the gait, but for you suggestion we change the sentence. Line 88

“All serous exudate was observed over the coronary bands in front limbs, but had a normal gait”

Point 8: 99: what were the signs associated with uveitis?

Response 8: Dear reviewer, according the suggestion of point 8 we added the signs observed

Changes in lines 99-100

“Both eyes were with bilateral corneal ulcers and uveitis (Figure 1B), with signs such as ephiphora, blepharospasm, corneal edema and episcleral congestion”

Point 9: 121: was endoscopy of the trachea performed? If so, what was seen?

Response 9: Dear reviewer you are correct. About the comment of point 9, we added more information.

Paragraph 122-125, we added the underlined

Airway endoscopy was performed the second day (1800PVS9150, Portascope, Fl, United State) and mild to moderate irritation with hyperemia of the mucous membrane of common nasal meatus and the first part of ventral nasal meatus on both nostrils, also mild hyperemia in pharynx and trachea was observed.

Point 10: 321: a (very) brief comparison with barn fires could be made here.

Response 10:Dear reviewer according to point 10, we added your suggestion. Review lines 322-326. The changes is underlined

 Case report of horses injured in open range fires described one horse with problems by smoke inhalation which resolved in 24 h [5]. Barn fires are unfortunately too common, and each year, hundreds of horses die or are severely injured in these incidents. Gimenez et al. describe a review of strategies to prevent and respond to barn fires in the horse industry [22].

Point 11: 333: and reported for horses (reference 5)

Response 11: According the comment of point 11, it was added your suggestion, thanks for the observation

Review line 338

Point 12: 144: where (on the body) was the edema present?

Response 12: Dear reviewer, according the comment of point 12. Please read the line 95. And also I added  the word “described “

Line 93-95 

Most of the affected areas were classified as deep second-degree burns with some small areas of third-degree burns on the distal area of front limbs and face. There was also facial, abdominal and limb edema.

Now in line 145

“Edema described was observed………..”

Point 13: 204: Presumably the horizontal line moved further down the hoof over time. Please clarify (as is done briefly in the discussion)

Response 13: Dear reviewer the comment of point 13 was made in cases D1 and D2. To clarify we write en discussion the next Lines 376-377

“However, there was abnormal hoof growth, moved down over time until the old hoof was replaced, observed which was noted 8 month after the fire”
